# Yifei Tongluo, a Chinese Herbal Formula, Suppresses Tumor Growth and Metastasis and Exerts Immunomodulatory Effect in Lewis Lung Carcinoma Mice

**DOI:** 10.3390/molecules24040731

**Published:** 2019-02-18

**Authors:** Qiuchen Qi, Yanhong Hou, Ang Li, Yueyue Sun, Siying Li, Zhongxi Zhao

**Affiliations:** 1School of Pharmaceutical Sciences, Shandong University, 44 West Wenhua Road, Jinan 250012, Shandong, China; qiuchensd@163.com (Q.Q.); houyanhong3908@163.com (Y.H.); liangliang0725@aliyun.com (A.L.); 13021717075@163.com (Y.S.); 2Department of Pathology and Pathophysiology, School of Basic Medical Sciences, Shandong University, 44 West Wenhua Road, Jinan 250012, China; li-siying@hotmail.com; 3Shandong Provincial Key Laboratory of Mucosal and Transdermal Drug Delivery Technologies, Shandong Academy of Pharmaceutical Sciences, 989 Xinluo Street, Jinan 250101, Shandong, China; 4Shandong Engineering & Technology Research Center for Jujube Food and Drug, 44 West Wenhua Road, Jinan 250012, Shandong, China

**Keywords:** lung cancer, antitumor, anti-metastasis, apoptosis, MMPs, immunomodulatory, MAPK pathway

## Abstract

This study was aimed to investigate the anti-tumor, anti-metastasis and immunomodulatory effects of Yifei Tongluo (YFTL), a Chinese herbal formula, in Lewis lung carcinoma mice and to explore the underlying mechanisms. LLC cells were inoculated subcutaneously in C57BL/6 mice to establish the Lewis lung carcinoma model. We observed that YFTL effectively inhibited tumor growth and prolonged the overall survival of tumor-bearing mice. Additionally, YFTL treatment resulted in a significantly decreased number of surface lung metastatic lesions compared with the model control group. Meanwhile, TUNEL staining confirmed that the tumors from YFTL-treated mice exhibited a markedly higher apoptotic index. The results suggest that Akt and mitogen-activated protein kinase (MAPKs) pathways may be involved in YFTL-induced apoptosis. The results show that YFTL also inhibited the vascular endothelial growth factor (VEGF), matrix metalloproteinases (MMP)-2, MMP-9, N-cadherin, and Vimentin expression, but increased E-cadherin expression. Mechanistic studies indicated that YFTL could suppress the angiogenesis and the epithelial-mesenchymal transition (EMT) of the tumor through Akt/ERK1/2 and TGFβ1/Smad2 pathways. In addition, YFTL also showed immunomodulatory activities in improving the immunosuppressive state of tumor-bearing mice. Therefore, our findings could support the development of YFTL as a potential antineoplastic agent and a potentially useful anti-metastatic agent for lung carcinoma therapy.

## 1. Introduction

Lung cancer is one of the most common causes of cancer death worldwide with high incidence and mortality among all types of cancer. Non-small cell lung cancer (NSCLC) is a main type of all lung cancers, accounting for nearly 85% [1,2]. Despite advances in conventional treatments including surgery, radiotherapy, and chemotherapy in the last few decades, the side effects and drug-resistance lead to poor quality of life and survival outcomes, which already have been the bottleneck of cancer treatment [3,4,5]. The major reasons for treatment failure and poor prognosis in lung cancer include tumor burden, invasion, and metastasis [6]. Therefore, it is urgent and necessary to develop new drugs with high efficiency, low toxicity and multi-target for treatment of lung cancer.

Metastasis is a critical step in cancer progression and is the leading cause of cancer-related deaths in human. Tumor metastasis is a complex multistep process, including tumor angiogenesis, cell migration, cell attachment, degradation of extracellular matrix (ECM) and invasion [7]. Angiogenesis is a basic biological characteristic of the tumor and it has been a promising therapeutic strategy for inhibiting the tumor growth by interfering with the binding of VEGF to VEGFR [8,9,10]. MMP-2 and MMP-9 are highly expressed in malignant tumors, which degrade an array of ECM proteins, and have a profound impact on cancer progression by influencing angiogenesis, invasion, and migration [11].The EMT plays a crucial role in lung cancer metastasis, which also causes poor prognosis and drug-resistance [12,13]. The EMT process can be modulated by multiple signaling pathways, such as TGF-β1, MAPK and PI3K/AKT during malignant tumor progression [12,14]. These suggest that the inhibition of angiogenesis and suppression of the activation of EMT may be effective for the treatment of cancer.

Traditional Chinese medicine (TCM) has been applied to treat lung cancer with a large number of advantages, including a wide range of sources, low costs, fewer side effects, and multiple targets over a long history. Additionally, it has advantages in preventing tumor recurrence and metastasis compared with radiotherapy and chemotherapy [15,16,17]. Therefore, development and application of TCM has already become a hotspot for cancer studies. The prescription Yifei Tongluo (YFTL) evaluated in this work was composed of eleven Chinese medicinal herbs (Table 1). It has been commonly used to treat lung-related diseases such as multidrug resistant tuberculosis in China and it can improve the immune activity and modulate the Th1/Th2 immune responses [18,19,20,21]. The major chemical components of YFTL include coumarins, phenols, flavonoids, alkaloids, and malate esters [22]. Although YFTL is not traditionally used to treat for lung cancer, some of its active components, such as Chlorogenic acid [23], Rutin [24,25], linarin [26,27], and esculetin [28,29] have been identified for their effective anti-tumor and anti-metastasis effects in different models. According to the previous reports, the inhibition on tumor growth and metastasis might be associated with the inhibition of cell proliferation and induction of apoptosis, as well as suppression of angiogenesis, alleviation of inflammation responses, up-regulation of E-cadherin and down-regulation of MMP-2 and MMP-9 levels [27,30,31].

Whether YFTL exerts inhibitory effects in tumor progression and the underlying mechanisms of action have not been elucidated. In the present study, we aimed to determine the anticancer effect of YFTL in Lewis pulmonary adenoma mice by monitoring its effect on tumor growth, survival, cell proliferation, apoptosis, angiogenesis, MMPs, EMT, and immune activity. Additonally, we explored the mechanisms underlying its action against Lewis lung cancer.

## 2. Results

### 2.1. Tumor Growth and Survival in Lewis Lung Cancer-Bearing Mice

The inhibitory effect of YFTL was evaluated in Lewis lung carcinoma mice. At the end of the experiment, tumor specimens were isolated and weighted. As shown in Figure 1A,B, the average tumor weights in the two YFTL groups were significantly lower than that in the model control group (MC). At doses of 200 mg/kg and 400 mg/kg (YFTL-L and YFTL-H), YFTL, administrated intragastrically, suppressed the tumor growth by 48.35% and 61.99% (*p* < 0.001), respectively. In addition, there was no significant change in the body weights after YFTL treatment (Figure 1D), and biochemical serum analyses of alanine transaminase (ALT) and aspartate transaminase (AST) indicated no obvious effects on liver functions in the YFTL-treated mice during the treatment period. The results of the survival assay (Figure 1E) showed that YFTL administration prolonged the survival of tumor-bearing mice. There were two of eight mice in the MC group that lived to day 40 and all of them died by day 42, whereas 80% of mice in the YFTL-L group and all mice in the YFTL-H group lived to day 40. The median survival time of YFTL-L (42 days) and YFTL-H group (48 days) was significantly longer than that in the MC group (36 days). These results indicate that YFTL treatment obviously inhibited tumor growth and prolonged survival in Lewis lung cancer-bearing mice.

### 2.2. Anti-Metastasis Effect of YFTL in Lewis Lung Cancer-Bearing Mice

The anti-metastatic efficacy of YFTL was evaluated by counting the number of nodules on the lung surface and by histological HE staining. The results showed that many metastatic nodules formed in the lungs in the model control group, whereas mice treated with YFTL (YFTL-L, YFTL-H) had fewer nodules on the lung surface (Figure 1F,G). Furthermore, HE staining of the lung tissue showed that YFTL alleviated infiltration of inflammatory cells and alveolar septal thickening in the lung tissue of LLC tumor-bearing mice, demonstrating a potent anti-metastasis efficacy of YFTL.

### 2.3. Effect of YFTL on the Proliferation and Apoptosis in Lewis Tumor-Bearing Mice

HE staining showed a remarkably increased necrosis in the tumors of YFTL treatment groups compared to the model control group (Figure 1C). Uncontrolled tumor cell proliferation is a characteristic feature of most cancers. Therefore, we detected the cell proliferation markers (Ki67 and PCNA) to analyze the anti-proliferation effect of YFTL on the Lewis lung tumor. As shown in Figure 2A–C, YFTL treatment obviously reduced the expression of Ki-67 (*p* < 0.01) and PCNA (*p* < 0.05). In addition, to determine whether cell apoptosis is involved in the tumor growth inhibitory effects of YFTL, we elucidated the apoptosis of tumor tissues by TUNEL assay and IHC analysis. The TUNEL assay revealed an increase in the number of apoptotic cells in the YFTL treatment groups (YFTL-L and YFTL-H) (*p* < 0.05 or *p* < 0.001) compared with the MC group (Figure 2A,D). The p53 expression was detected by IHC analysis which played an important role in the regulation of cell cycle and apoptosis. The expression of p53 (*p* < 0.05) in the Lewis tumor of the YFTL groups was distinctly higher than that of the MC group. Additonally, Western blot analysis of apoptotic markers showed that YFTL up-regulated Bax expression (Figure 2F,G). Taken together, the above results showed that YFTL inhibited proliferation and induced apoptosis in Lewis lung cancer mice.

### 2.4. Effect of YFTL on Angiogenesis in Lewis Tumor-Bearing Mice

To illustrate the anti-metastatic mechanism of YFTL, tumor sections were stained with anti-CD34 antibody to detect the microvessel density (MVD). And we detected VEGF immunochemistry staining in tumor tissues and measured the VEGF level in serum to evaluate the angiogenesis expression. As a result (Figure 3), the density of anti-CD34 positive microvessels (brown color) in the experimental groups was lower than in the model control group. YFTL had prominent inhibitory effectiveness on VEGF expression not only in tumor tissues but also in serum (*p* < 0.05 or *p* < 0.001). These data indicated that YFTL markedly inhibited tumor angiogenesis compared with MC group in Lewis lung cancer-bearing mice.

### 2.5. Effect of YFTL on EMT and MMPs in Tumor and Lung Tissues of Lewis Lung Cancer Mice

The epithelial-mesenchymal transition (EMT) and mass production of MMPs are pivotal factors for tumor invasion and metastasis [12]. MMP-2 and MMP-9 are the most important enzymes to degrade the extracellular matrix [32]. EMT is defined as switching of polarized epithelial cells to a migratory fibroblast phenotype. The characteristics of EMT include the loss of EMT-related protein expression, such as E-cadherin, and the up-regulation of interstitial related protein expression, including N-cadherin and Vimentin [33]. To illustrate the anti-metastatic mechanism of YFTL, we measured E-cadherin, N-cadherin, Vimentin, MMP-2, and MMP-9 expression in tumor and lung tissues of Lewis lung cancer mice by immunohistochemistry and Western blot analysis. As shown in Figure 4A, YFTL significantly reduced MMP-2, MMP-9, N-cadherin and Vimentin expression, but markedly increased E-cadherin expression in the tumor and lung tissues of tumor-bearing mice compared with the MC group. Western blot analysis revealed that the expression levels of the protein MMP-9, MMP-2 and Vimentin in tumor tissues of YFTL treatment groups were noticeably lower than that in the MC group (Figure 4B,C). Taken together, the above results showed that YFTL evidently inhibited EMT in Lewis lung cancer mice.

### 2.6. Effect of YFTL on Splenocytes Proliferation and NK Cell Activity in Lewis Tumor-Bearing Mice

We evaluated T and B lymphocyte proliferation stimulated by Con A and LPS, respectively. As shown in Figure 5A,B, YFTL administration at doses of 200 mg/kg or 400 mg/kg dramatically promoted splenocytes restoration compared to the MC group (*p* < 0.05 or *p* < 0.01). YFTL had a stronger accelerative effect on T lymphocytes restoration. NK cell is a major population of cytotoxic lymphocyte and plays an important role in the defense against tumors [34]. In this study, administrated with YFTL significantly increased the NK cytotoxic activity at a high dose of YFTL (YFTL-H) in Lewis lung cancer-bearing mice (*p* < 0.05) (Figure 5C).

### 2.7. Effect of YFTL on CD4^+^ T Cell, CD8^+^ T Cell, and NK Cell Subsets in Lewis Tumor Mice

CD4^+^ and CD8^+^ T lymphocytes are the most important immune cells in the regulating component of immune system respectively via releasing pro-inflammatory cytokines or direct cytotoxic effects [35,36]. NK cell plays an important role in anti-tumor immunity. The results of the flow cytometric analysis demonstrated that the percentages of CD4^+^ and CD8^+^ T cells among total lymphocytes were dramatically higher than those in the MC group after the treatment of YFTL (400 mg/kg). In addition, YFTL (400 mg/kg) significantly increased the percentage of splenic NK cell (Figure 5D,E). Taken together, the above results show that the tumor suppression effect of YFTL in Lewis tumor-bearing mice may be associated with the up-regulation ratio of CD4^+^ T cell, CD8^+^ T cell, and NK cell subsets.

### 2.8. Effect of YFTL on Cytokines Levels in the Serum of Lewis Tumor-Bearing Mice

Based on the effect of YFTL on splenic lymphocytes and NK cell, the cytokine levels of IL-2, IFN-γ, IL-10, and TGF-β1 in the serum of tumor-bearing mice were detected. As shown in Figure 6, We found that YFTL treatment obviously increased the serum Th1-type cytokines levels of IFN-γ (*p* < 0.01) and IL-2 (*p* < 0.01) at doses of 200 mg/kg and 400 mg/kg (YFTL-L and YFTL-H). In contrast, YFTL treatment caused a significant reduction in IL-10 (*p* < 0.05) and TGF-β1 (*p* < 0.01) in the serum compared to the MC group.

### 2.9. Regulation of PI3K/AKT, MAPK, and TGFβ/Smad2 Pathways in YFTL-Treated Lewis Tumor-Bearing Mice

According to the above results, YFTL inhibited the proliferation, tumor angiogenesis and EMT, and induced apoptosis in Lewis lung cancer mice. Additionally, YFTL exhibited immunomodulatory activities in improving the immunosuppression of tumor-bearing mice. To further investigate the regulation of major pathways by YFTL, the signaling proteins of the phosphatidylinositol-3-kinase (PI3K)/Akt, mitogen-activated protein kinase (MAPK) and TGFβ/Smad2 pathways were detected by Western blot analysis of tumor tissues. As a result, YFTL significantly inhibited the phosphorylation of Akt, Erk1/2 and Smad2, and decreased the expression of TGFβ in Lewis tumor tissues. We found that YFTL also activated the phosphorylation of p38 MAPK and JNK (Figure 7). These data suggested that YFTL down-regulated PI3K/AKT, ERK1/2 and TGFβ1/Smad2 pathways and up-regulated JNK and p38 pathways to prevent cancer growth and tumor metastasis in Lewis tumor-bearing mice.

## 3. Discussion

YFTL has a long history of being utilized for the treatment of lung-related diseases in clinical, such as tuberculosis [22]. However, its therapeutic potential for lung cancer has not been well investigated. In the present study, we for the first time demonstrated that YFTL could inhibit the tumor growth and tumor metastasis, and prolonged the survival of tumor-bearing mice. Moreover, our study indicated that YFTL possessed immunomodulatory activity in tumor-bearing mice. The present study showed that the antitumor effect of YFTL in mice was associated closely with induction of apoptosis, suppression of angiogenesis and inhibition of MMPs and EMT.

It has been reported that major active components of YFTL such as esculetin-inhibited cancer cell growth by inhibiting cell proliferation and inducing apoptosis [28,29]. In the study, YFTL induced apoptosis as indicated by TUNEL staining in tumor tissues and reduced the expression of two proliferation markers, PCNA and Ki-67. In addition, p53 expression was elevated by YFTL treatment [37]. Previous results indicate that AKT/MAPK are important signaling pathways regulating proliferation and apoptosis [38]. It has been demonstrated that AKT pathway is aberrantly activated in many human cancers and plays a central role in apoptosis inhibition through regulating p53 [39] Our data revealed that YFTL suppressed tumor proliferation and induced apoptosis via activation of the p38MAPK and JNK pathway, and down-regulating Akt pathways [40].

Invasion and metastasis are the main factors leading to treatment failure and poor prognosis of lung cancer. Angiogenesis, MMPs and EMT play crucial roles in the process of tumor metastasis, which have long been drug targets [41,42]. Chlorogenic acid, major active component of YFTL, significantly suppresses the proliferation of NSCLC and inhibits the activity of MMP-2 [23]. Additionally, linarin, a major active component of YFTL, inhibited tumor growth and metastasis via down-regulation of Akt and repressed the MMP-9-dependent invasion pathway [26,27]. Lin et al., demonstrated that curcumin had the anti-tumor and anti-angiogenic potential by the suppression of VEGF in non-small lung cancer line, A549 [43]. Our study showed that YFTL treatment down-regulated the activities of MMP-2 and -9, decreased the VEGF expression and inhibited the EMT in Lewis lung cancer-bearing mice, implicating that YFTL has the anti-metastatic and anti-angiogenic potential.

Previous results indicate that AKT/ERK are also critical signaling pathways regulating migration and invasion of cancer cells during carcinogenesis [38]. MMP2/9 are deemed to promote metastasis activated by PI3K/Akt pathway, and the MMPs expression is also regulated by ERK pathway [44]. Our results illustrated that YFTL exhibited anti-metastasis effect on Lewis lung carcinomas by attenuating the activities of MMP-2 and -9 via down-regulating Akt/ERK pathways [45]. In addition, TGF-β1 signaling plays a major role in regulating several different biological processes involving cell-growth, apoptosis, angiogenesis, and EMT [46]. TGF-β1 can induce EMT via Smad2 or via a Smad-independent pathway. Our results demonstrate that YFTL inhibited EMT by suppression of TGF-β1 signaling pathway [47]. Consistent with these reports, our results illustrated that the suppression of Akt/ERK1/2 and TGF-β1/Smad2 pathways by YFTL might contribute to the inhibition of EMT.

Recently, several studies imply that the TCM exerts anti-tumor effect mainly through regulating the immune system [15]. Our data showed that the antitumor activity of YFTL may be related to the activation of the immune response of the host organism by the stimulation of NK cells, T cells, and modulate the cytokines production [48]. Serum cytokines are important in immune system and play a vital role in mediating the host defense. The IL-2 can promote the proliferation of responsive T cells and IFN-γ play important roles in depressing tumor cell growth and inducing apoptosis of tumor cells [49,50]. In the present experiment, YFTL could significantly increase the two cytokines at dose of 200 and 400 mg/kg compared with the model control group. The data implied that YFTL might exert anti-tumor effect by downregulating immunosuppressive molecules such as TGF-β1 and IL-10, and upregulating the level of IFN-ϒ and IL-2. CD4^+^ T cells, CD8^+^ T cells, and NK cells play important roles in anti-tumor immunity [51,52]. Our results showed that YFTL could increase the percentages of the three cells and the up-regulated ratios of the CD4^+^ T cells, CD8^+^ T cells, and NK cells might participate in the anti-tumor activity of YFTL in vivo. MAPKs are important mediators of cytokine expressions and are critically involved in the immune response [53]. These immunomodulatory activities induced by YFTL may be achieved through the MAPK pathway.

TCM, as an important part of complementary and alternative therapy, is often used for maintenance therapy of lung cancer mainly due to its low toxicity or adverse effects, less cost, and better improvement in clinical outcomes [54]. It is well-known that herbal formula therapy is one of the most important characteristics of TCM [55]. YFTL formula contains a variety of active ingredients with a high structural diversity, which can synergistically enhance the therapeutic effects such as anti-proliferation, anti-resistance and immuno-modulation. It is found that long-term using of chemotherapeutic drugs lead to drug resistance, thus generating more toxic and side effects including immunosuppression [56]. Many first-line chemotherapy drugs for non-small cell lung cancer, such as vinorelbine are often limited for clinical use due to their side effects. There are more and more clinical studies show that the combination TCM with vinorelbine has a synergistic antitumor effect, which can improve quality of life and immunity of patients, bone marrow suppression, and so on [57]. YFTL contains multiple herbs for balance of Yin and Yang, a system of two opposites but complementary aspects of nature to maintain immune homeostasis [58]. Therefore, it is promising to combine YFTL with the chemotherapy drugs to reduce side effects and improve efficacy. In addition, multiple changes in tumor micro-environments, as well as emergent immunogenic mechanisms, enable resistance to immune checkpoint inhibitors. For this reason, the single-agent anti PD-1/PD-L1 is of limited clinical benefits to patients. According to the basic theory of TCM, the concept of holism in the herbal formula is consistent with the tumor microenvironment just as macroscopic and microscopic aspects [33]. Therefore, the combination YFTL with immune checkpoint inhibitors might be promising in cancer therapeutics.

In summary, we for the first time provide evidence that YFTL can effectively inhibit tumor growth and metastasis, and improve the immunosuppressive state in Lewis lung cancer mice. The underlying mechanisms of its antitumor and anti-metastasis effects might involve induction of apoptosis, suppression of angiogenesis and inhibition of EMT by regulating AKT/MAPK and TGF-β1/Smad2 pathways. Therefore, our study provided scientific evidence for the potential therapeutic value of YFTL in the treatment of lung cancer.

## 4. Materials and Methods

### 4.1. Materials and Reagents

YFTL was obtained from Changsha Central Hospital (Hunan, China). The HE staining kit was purchased from Beyotime (Jiangsu, China). Antibodies against proliferating cell nuclear antigen (PCNA), p53, MMP-2, MMP-9, E-cadherin, N-cadherin vimentin, VEGF, TGFβ1, Smad2, and p-Smad2 were purchased from Abcam (Cambridge, UK). The cytokine ELISA kits were purchased from Lianke Biotechnology Co. Ltd. (Hangzhou, China). Anti-CD3^+^ (FITC), anti-CD4^+^ (PE-Cy5), anti-CD8^+^ (PE) and anti-NK1.1(PE-Cy7) antibodies were provided by Bio Legend, Inc. (San Diego, CA, USA). Anti-AKT and anti-p-AKT antibodies were provided by Santa Cruz Biotechnology (Santa Cruz, CA, USA). Antibodies against ERK1/2, p-ERK1/2, p38, p-p38, JNK, and p-JNK were from Cell Signaling Technologies (Danvers, MA, USA). Anti-β-actin antibodies were from Proteintech Biotechnology (Rocky Hill, USA). Other reagents were purchased from Sigma-Aldrich (St. Louis, Missouri, USA).

### 4.2. Cell Culture

The mouse lung adenocarcinoma cells line LLC was obtained from Cell Bank of China (Shanghai, China). Cancer cells were maintained in DMEM medium containing 10% fetal bovine serum, 100 U/mL of penicillin and 100 mg/mL of streptomycin in a humidified incubator under 5% CO2 at 37 °C.

### 4.3. Animals

Male C57BL/6 mice (6–8 weeks old, 18–20 g) were obtained from Huafukang Biological Products Co. Ltd. (Beijing, China). Mice were kept in polypropylene cages in the laboratory’s animal room at standard temperature (25 ± 2 °C) with relative humidity of 60 ± 15%, and 12 h light-dark cycle. Animals were given food and water freely. During housing, animals were monitored twice daily for health status. No adverse events were observed. All the experimental procedures were approved by the guidelines of the Ethical Committee Experimental Animal Center of Shandong University (no. 2016020, Jinan, China).

### 4.4. Antitumor and Anti-Metastasis Efficacy in Lewis Lung Carcinoma Mice

After one week of acclimatization, a subcutaneous injection of LLC cells 5 × 10^5^ suspended in 0.2 mL PBS was implanted into the right flank of each C57BL/6 mouse. Starting the next day, the mice were randomly divided equally into three groups (18 mice each group) as follows: the model control group (MC) (received the 0.5% sodium carboxymethyl cellulose solution by gavage); YFTL-L group (received 200 mg/kg/day YFTL by gavage); YFTL-H group (received 400 mg/kg/day YFTL by gavage). YFTL was dissolved in the 0.5% sodium carboxymethyl cellulose solution. The treatments were administered consistently at 11 am every day. In each group, 10 mice were used for tumor growth and another eight mice were used for survival analysis. The mouse body weights were recorded before the daily administration during the study period. In the tumor growth experiment, the mice were treated day for 21 days and none of the mice died. At the end of the study, the animals were sacrificed by cervical dislocation following anesthesia using isoflurane. All efforts were made to minimize animal suffering and to reduce the number of animals used. Additionally, samples (serum, tumors and major organs) were immediately removed and frozen in liquid nitrogen. The spleen tissues were harvested to detect the immunomodulatory effect of YFTL. The weights of tumor tissues were measured. Part of tumor specimens were exposed to 4% paraformaldehyde fixation for hematoxylin-eosin (HE) staining and immunohistochemistry study, and the remaining tissues were kept in −80 °C for western blot analysis. The lungs were inflated with 30% sucrose, fixed in Bouin’s solution for 18 h, and stored in 70% ethanol. Each of the five pulmonary lobes was separated and surface tumors were counted under dissecting microscope. After counting, the lungs were sectioned and stained with HE.

### 4.5. H&E, TUNEL and Immunohistochemistry Assay

After isolated from mice, tumor and lung tissues were fixed in 4% paraformaldehyde, embedded in paraffin and cut into 4-μm sections. Then these sections were deparaffinized and stained with hematoxylin and eosin. Finally, the sections were dehydrated and mounted for imaging.

The type of cell death (necrosis/apoptosis) was evaluated by terminal deoxynucleotidyl transferase (TdT)-mediated dUTP nick end-labeling (TUNEL) assay with an apoptosis detection kit (Roche, California, USA). Briefly, after deparaffination and permeabilisation, tumor sections were incubated with TUNEL reaction mixture for 60 min and Converter-POD solution for 30 min at 37 °C in dark. The sections were visualized by exposing to diaminobenzidine (DAB) substrate (Zhongshan Jinqiao Corp. Beijing, China) and counterstained with hematoxylin (Solarbio Biotechnology Corp. Beijing, China).

For immunohistochemical analysis, the tumor sections and lung sections (4 μm thick) were blocked and incubated with anti-Ki67, anti-p53 (1:200), anti-CD34, anti-VEGF (1:50), anti-E-cadherin (1:300), anti-N-cadherin (1:300), anti-vimentin (1:500), anti-MMP-2 (1:100), or anti-MMP-9 (1:200) antibodies overnight in 4 °C. Subsequently, immunostaining was performed according to the standard protocol using a DAB Substrate Kit. The sections were counterstained with hematoxylin and analyzed under an Olympus microscope (Tokyo, Japan). The quantitative analysis of IHC was measured in 5 random fields per tumor by light microscopy at a magnification of 100× and quantitated by Image-Pro Plus 6.0 (Media Cybernetics, Bethesda, MA, USA).

### 4.6. Determination of Cytokines and VEGF in the Serum by ELISA

Serum samples were prepared by centrifuging the whole blood at 3500 rpm at 4 °C for 10 min. The levels of VEGF and cytokines for IFN-γ, IL-2, TGF-β1, and IL-10 in the serum were measured by commercially available ELISA kits (MultiSciences Biotech Co., Ltd., Hangzhou, China) according to the manufacturer’s instructions. The analysis of aspartate aminotransferases (AST) and alanine aminotransferase (ALT) in the serum was performed using commercial kits (Nanjing Jiancheng Bioengineering Institute, China).

### 4.7. Measurement of Lymphocyte Proliferation

Splenic lymphocytes were separated from mouse spleens in order to analyze the lymphocyte proliferation using a CCK8 assay as described previously [22]. Briefly, the extirpated spleens were treated in germ-free condition. Single-cell spleen suspensions were pooled in serum-free RPMI-1640 medium by filtering the suspension through sieve mesh with the aid of a glass homogenizer to exert gentle pressure on the spleen fragments. The cells were treated with the lysis buffer (0.747% Tris-NH4Cl buffer, pH7.4) to get rid of red blood cells, while the remaining cells were centrifuged at 200× *g* for 5 min, followed by washing with cold PBS for three times. Then single splenocyte was resuspended in RPMI-1640 and adjusted to the concentration of 2 × 10^6^ cells/mL. The splenocytes (100 μL/well) were seeded into the 96-well plates with or without Con A (5 μg/mL) or LPS (10 μg/mL) and incubated at 37 °C in a humidified atmosphere of 5% CO2 for 70 h. Lymphocyte proliferation was estimated based on the method of CCK-8. Briefly, 10 μL of WST-8 (5 mg/mL) was added to each well incubated for another 2 h and then the absorbance was detected at 450 nm using a microplate reader.

### 4.8. Splenic NK Cell Cytotoxicity Activity Assays

Splenic lymphocytes obtained from the spleen were used as effector cells for splenic NK cell activity assay as described above [59] YAC-1 cell was used as the target cell for NK cell. Briefly, the effector cells (5 × 10^5^ cells/well) in the 96-well plates were co-cultured with target cells (1 × 10^4^ cells/well) at a ratio of effector to target cells of 50:1. The plates were then incubated for 22 h at 37 °C in 5% CO_2_ atmosphere. The cytotoxic activity of NK cell was measured based on the method of CCK-8 and calculated according to the following formula:(1)$$\mathrm{Cytotoxicity}\;\left(\%\right)=\frac{T_{\mathrm{OD}}-\left(S_{\mathrm{OD}}-E_{\mathrm{OD}}\right)}{T_{\mathrm{OD}}}\times 100\%,$$
where T_OD_ is an optical density value of the target cell control, S_OD_ is an optical density value of test samples, and E_OD_ is an optical density value of the effector cell control.

### 4.9. Determination of Spleen Lymphocyte Phenotype

The spleen lymphocytes were prepared and adjusted to 1 × 10^7^/mL and incubated with anti-CD3^+^ (FITC), anti-CD4^+^ (PE-Cy5), anti-CD8^+^ (PE) and anti-NK1.1 (PE-Cy7) (1:200) for 30 min at 4 °C under dark. The spleen lymphocytes were washed for three times with cell staining buffer. Then the percentages of CD4^+^ T cell, CD8^+^ T cell and NK cell were analyzed by flow cytometry (Beckman Coulter FC500, Miami, FL, USA).

### 4.10. Western Blot Analysis

Tumor tissues were lysed in RIPA buffer containing 50 mM Tris-HCl (pH 7.4), 150 mM NaCl, 1% NP-40, 0.5% sodium deoxycholate, 2 mM sodium fluoride, 2 mM EDTA, 0.1% SDS, and PMSF. The supernatant liquids were then collected and the total proteins were determined by a BCA kit (Beyotime Institute of Biotechnology, Beijing, China). The lysates were separated using 10% SDS-PAGE and electro-transferred onto the polyvinylidene difluoride (PVDF) membrane (Millipore Corp. Burlington, Massachusetts, USA). Following this, the membrane was exposed to primary antibodies (4 °C, 24 h) against MMP-2, MMP-9, vimentin, TGFβ1, Smad2, p-Smad2, AKT, p-AKT, ERK1/2, p-ERK1/2, p38, p-p38, JNK, and p-JNK. The PVDF membranes were washed in Tris buffered saline (TBS) containing 0.1% Tween-20 (TBST) and incubated with appropriate secondary antibodies for 1 h. Bound antibodies were visualized by an enhanced chemiluminescence reagent (Millipore) and the images were captured by Alphalmager HP system (Cell Biosciences, Inc., Santa Clara, Ca, USA).

### 4.11. Statistical Analysis

Data are showed as mean ± SD. The statistical significance of the differences between various groups was determined by either a Student’s t-test or an ANOVA analysis for multiple comparisons by Prism version 5.0 (GraphPad Software, Inc.). Values of *p* < 0.05 were considered statistically significant.

## Figures and Tables

**Figure 1 molecules-24-00731-f001:**
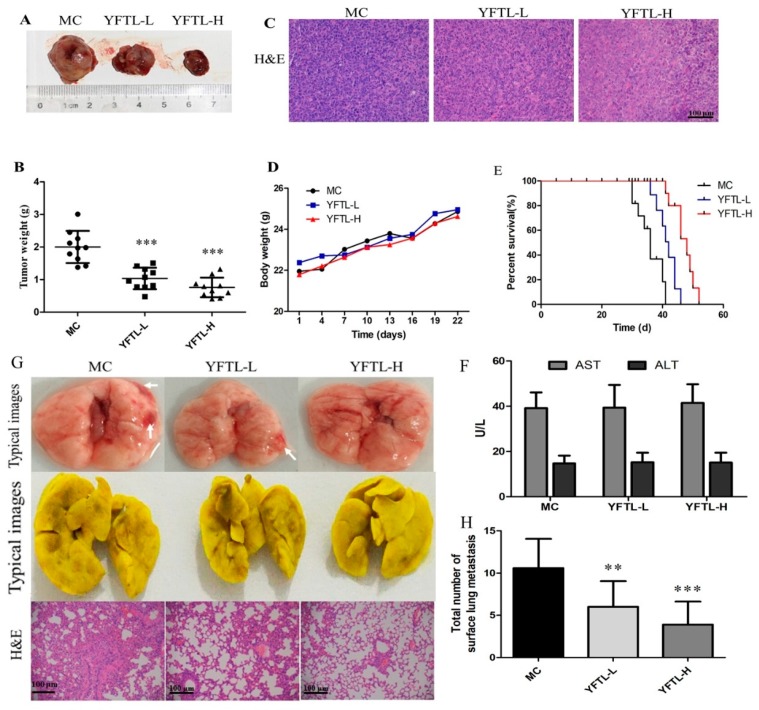
Antitumor and anti-metastasis effects of YFTL in Lewis lung cancer-bearing mice. (**A**) Tumors sampled from experiment groups; (**B**) tumor weight; (**C**) histological analysis of tumor sections stained with HE; (**D**) body weights during the treatment; **(E**) survival curves of mice in each group; (**F**) concentration of AST (U/L) and ALT (U/L) in serum of the mice shown as the mean ± SD. (**G**) Representative photographs of the whole lungs from mice in each group and HE staining of the lung tissue sections; and (**H**) the total number of nodules on the whole lung tissues. MC: model control group, treated with 0.5% sodium carboxymethyl cellulose solution by gavage; YFTL-L: treated with 200 mg/kg/day YFTL by gavage; YFTL-H: treated with 400 mg/kg/day YFTL by gavage. Data are expressed as the mean ± SD. ** *p* < 0.01 and *** *p* < 0.001 vs. the MC group.

**Figure 2 molecules-24-00731-f002:**
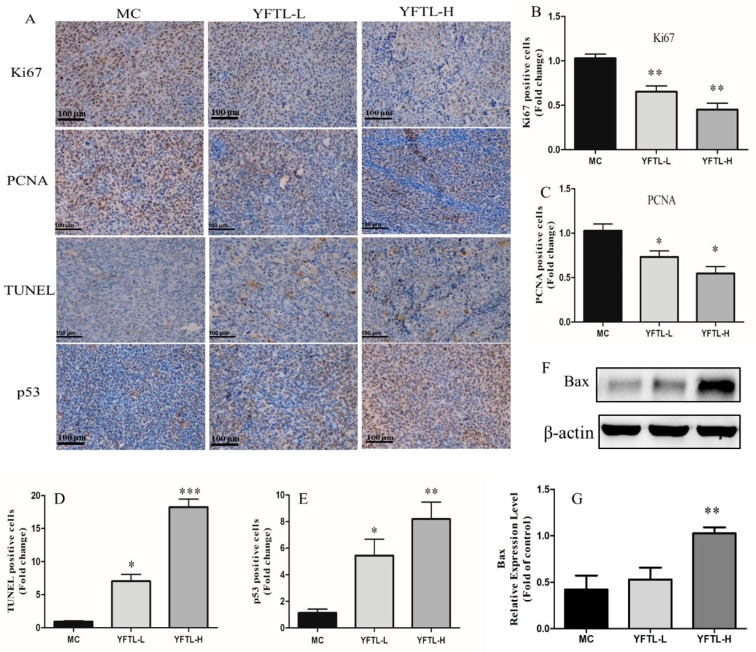
H&E, TUNEL and immunohistochemical assay of Lewis tumors. (**A**) Representative photomicrographs of H&E staining, TUNEL assay and immunohistochemistry of Ki67, PCNA, and p53 in tumor tissues. Scale bars are 100 μm. **(B)** Relative level of Ki-67 in tumor tissues. (**C**) Relative level of PCNA in tumor tissues. (**D**) Quantification of the numbers of TUNEL-positive cells. (**E**) Relative level of p53 in tumor tissues. (**F**) A representative band of Bax protein expression by Western blot analysis. **(G)** Relative protein level of Bax in tumor tissues. Data are expressed as a histogram of mean ± SD of three independent experiments, * *p* < 0.05, ** *p* < 0.01, and *** *p* < 0.001 vs. the MC group.

**Figure 3 molecules-24-00731-f003:**
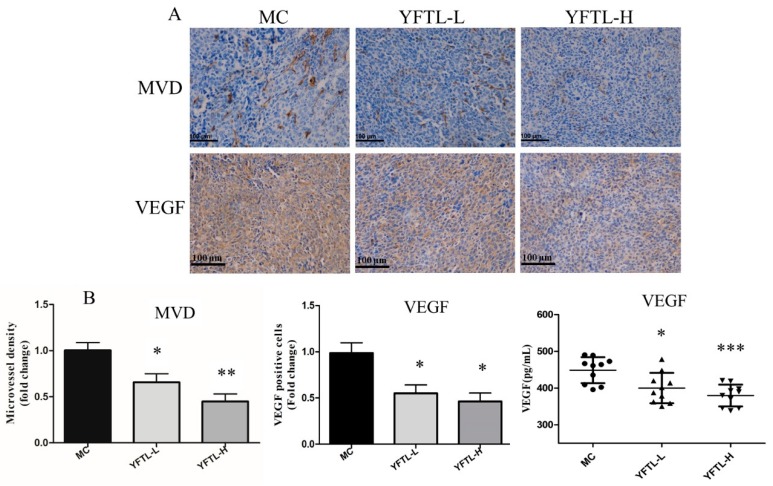
Effect of YFTL on angiogenesis in Lewis tumor-bearing mice. (**A**) Representative photomicrographs of CD34 and VEGF immunohistochemistry. Scale bars are 100 μm. (**B**) MVD was determined by blinded measurement of CD34 expression using IHC. Quantification of MVD and VEGF expression in tumor tissues and the VEGF level in serum. Data are expressed as mean ± SD, * *p* < 0.05, ** *p* < 0.01, and *** *p* < 0.001 vs. the MC group.

**Figure 4 molecules-24-00731-f004:**
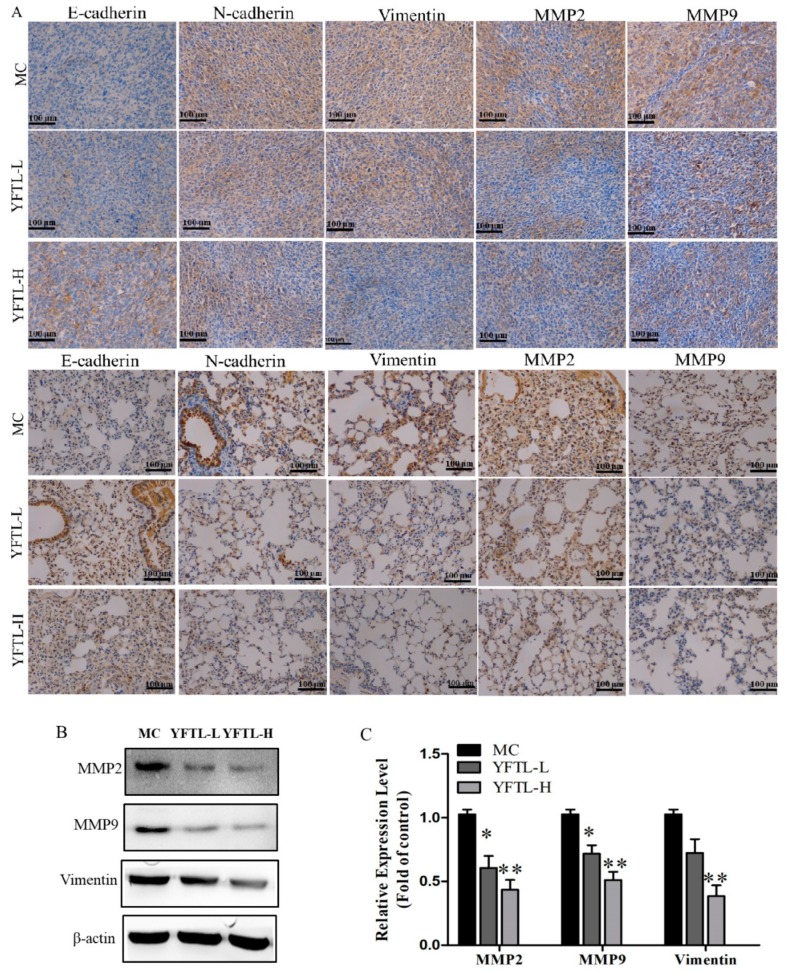
Effect of YFTL on EMT in tumor and lung tissues of Lewis lung cancer mice. (**A**) Representative photomicrographs of immunohistochemistry of MMP-2, MMP-9, E-cadherin, N-cadherin and Vimentin in tumor tissues and lung tissues. Scale bars are 100 μm. (**B**) A representative band of MMP-2, MMP-9, and Vimentin protein expression in tumor tissues by Western blot analysis. **(C)** Relative protein level of MMP-2, MMP-9, and Vimentin in tumor tissues. Data are expressed as a histogram of mean ± SD of three independent experiments, * *p* < 0.05, and ** *p* < 0.01 vs. the MC group.

**Figure 5 molecules-24-00731-f005:**
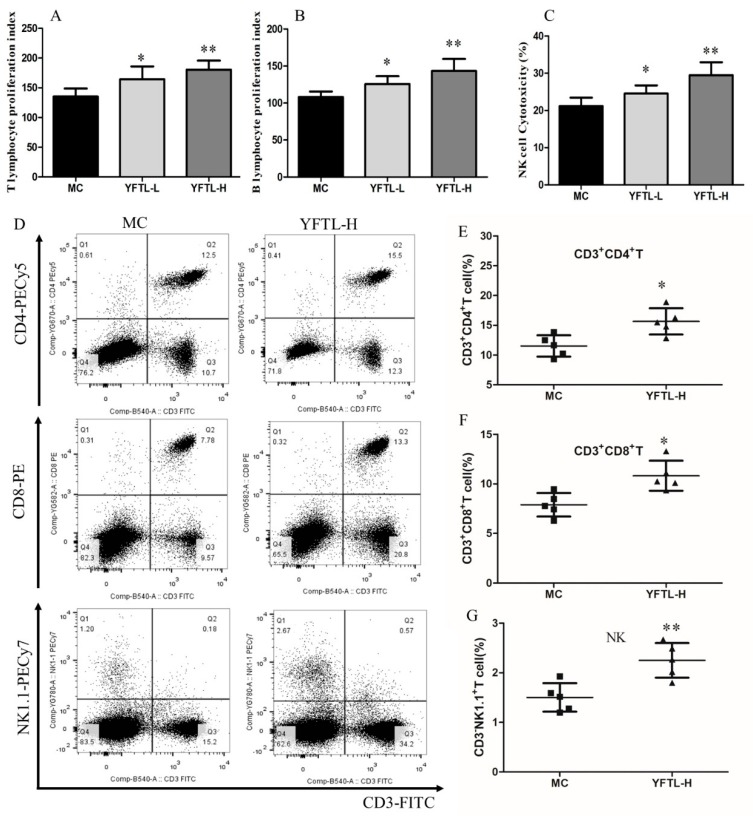
Effect of YFTL on splenocytes proliferation, NK cell activity and splenic lymphocyte subsets in Lewis tumor mice. (**A**) Splenocyte proliferation induced by Con A; (**B**) splenocyte proliferation induced by LPS; (**C**) splenic NK cytotoxicity; (**D**) splenic lymphocyte subsets detected by flow cytometry; (**E**) splenic lymphocyte subsets detected by flow cytometry; (**F**) CD4^+^ T cell, CD8^+^ T cell, and NK cell subsets analyzed by flow cytometry. Data are expressed as the mean ± SD, * *p* < 0.05, and ** *p* < 0.01 vs. the MC group.

**Figure 6 molecules-24-00731-f006:**
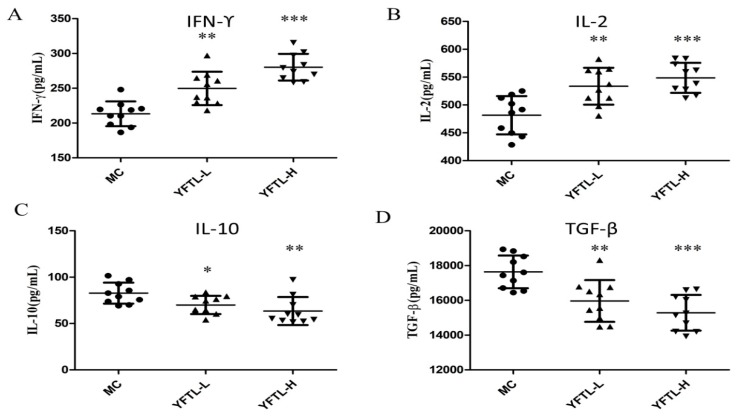
Effect of YFTL on cytokines production in the serum of LLC tumor-bearing mice. (**A**) IFN-γ; (**B**) IL-2; (**C**) IL-10; (**D**) TGF-β1. Data are expressed as mean ± SD. * *p* < 0.05, ** *p* < 0.01, and *** *p* < 0.001 vs. the MC group.

**Figure 7 molecules-24-00731-f007:**
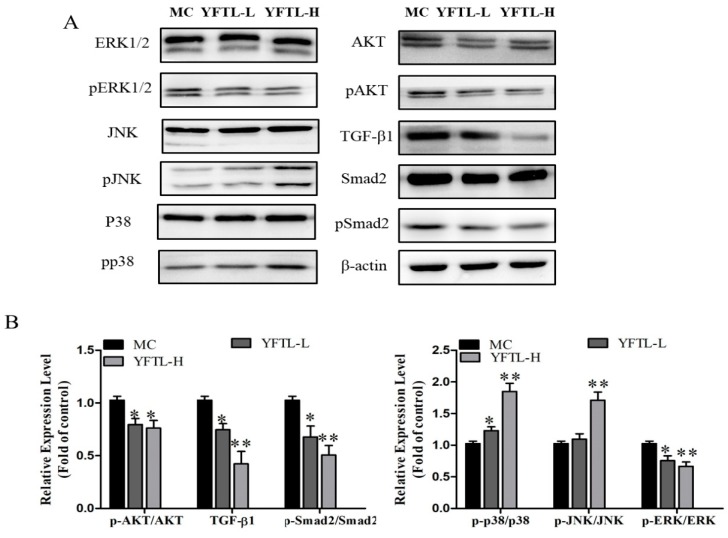
Regulation of PI3K/AKT, MAPK and TGFβ/Smad2 pathways in YFTL treated Lewis tumor-bearing mice. Data are expressed as a histogram of mean ± SD of three independent experiments. * *p* < 0.05 and ** *p* < 0.01 vs. the MC group.

**Table 1 molecules-24-00731-t001:** The information of each components in Yifei Tongluo granules.

Scientific Name	Herbal Name	Chinese Name	Family	Weight Ratio
*Polygonatum sibiricum* Red.	Polygonati Rhizoma	Huangjing	Liliaceae	2
*Bletilla sfriata* (Thunb.) Reiehb.f.	Bletillae Rhizoma	Baiji	Orchidaceae	1
*Stemonasessilifolia* (Miq.) Miq.	Stemonae Radix	Baibu	Stemonaceae	1.5
*Ardisiajaponica* (Thunb.) Blume	Ardisiae Japonicae Herba	Aidicha	Myrsinaceae	2
*Viola yedoensis* Makino	Violae Herba	Zihuadiding	Violaceae	2
*Tussilago farfara* L.	Farfarae Flos	Kuandonghua	Asteraceae	1
*Asparagus cochinchinensis* (Lour.) Merr.	Asparagi Radix	Tiandong	Liliaceae	1.5
*Cirsium japonicum* Fisch. ex DC.	Cirsii Japonica Herba	Daji	Asteraceae	1
*Pseudostellaria heterophylla* (Miq.) Pax ex Pax et Hoffm	Pseudostellariae Radix	Taizishen	Caryophyllaceae	1.5
*Luffa cylindrica* (L.) Roem	Luffae Fructus Retinervus	Sigualuo	Cucurbitaceae	1.5
*Trionyx sinensis* Wiegmann	Trionycis Carapax	Biejia	Trionychidae	1.5

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
