# Peer review of "Yifei Tongluo, a Chinese Herbal Formula, Suppresses Tumor Growth and Metastasis and Exerts Immunomodulatory Effect in Lewis Lung Carcinoma Mice"

_molecules, 2019, doi:10.3390/molecules24040731_

Round 1

Reviewer 1 Report

YFTL has been applied in targeting mycobacterial infection in clinical. The authors have demonstrated for the first time that YFTL inhibited tumor growth and metastasis in Lewis lung tumor xenograft model. Further mechanism analyses showed that YFTL suppressed tumor through multiple mechanisms, including inducing tumor cells apoptosis and activating immune system. It suggests YFTL is a promising drug in treating lung cancer. My concerns are as followings:

1.     Line 87, low and high doses of YFTL at 100 mg/kg and 200 mg/kg have been applied in this study. It is confusing that the doses are too high compared with previous publication showing that 2.8 mg/kg mouse YFTL (or 55.8 μg/mouse (20-g mouse) was used in a mouse model (Ref:  https://doi.org/10.1371/journal. pone.0203678 ).

2.     Line 89, a major side effect of YFTL is liver damage. Did the authors checked the levels of levels ALB, AST, ALT in serum upon treatment by YFTL?

3.     Fig. 1F, images showing the tumor nodules are not clear. Please change with new images. And how to quantify the tumor nodules in the lung?

4.     Fig. 3, could the author show the images of IHC staining for E-cadherin, N-Cadherin, Vimentin, MMP2, and MMP9 in the lung tumor nodules to show the EMT phenotype in metastatic lesions with and without YFTL treatment?

Author Response

Response to Reviewer 1 Comments:

Point 1: Line 87, low and high doses of YFTL at 100 mg/kg and 200 mg/kg have been applied in this study. It is confusing that the doses are too high compared with previous publication showing that 2.8 mg/kg mouse YFTL (or 55.8 μg/mouse (20-g mouse) was used in a mouse model (Ref: https://doi.org/10.1371/journal. pone.0203678).

Response 1: The doses of YFTL used in this study were determined based upon our preliminary studies, which was in the similar dose range of our previous study entitled “Immuno-enhancement effects of Yifei Tongluo Granules on cyclophosphamide- induced immunosuppression in Balb/c mice” (Ref: Journal of Ethnopharmacology 194 (2016) 72–82). The dosing formulation used in this work was the commercial dosage form (granules) and could be different from that in the cited study (Ref: https://doi.org/10.1371/journal. pone.0203678) due to the variations in the disease model and formulations used (i.e., pure herbal extracts vs. commercial dosage form containing excipients).

Point 2: Line 89, a major side effect of YFTL is liver damage. Did the authors checked the levels of levels ALB, AST, ALT in serum upon treatment by YFTL?

Response 2: We detected the levels of AST and ALT in serum, and as shown in Fig. 1F of the revised manuscript, biochemical serum analyses of alanine transaminase (ALT) and aspartate transaminase (AST) indicated no obvious effects on liver functions in the YFTL-treated mice during the treatment period.

Point 3: Fig. 1F, images showing the tumor nodules are not clear. Please change with new images. And how to quantify the tumor nodules in the lung?

Response 3: We have replaced the new images in Fig. 1F. According to the following references, the lungs were inflated with 30% sucrose, fixed in Bouin's solution for 18 h, and stored in 70% ethanol. Each of the five pulmonary lobes was separated and surface tumors were counted under a dissecting microscope.

1.     Wang, Y.; Xie, Y.; Li, J.; Peng, Z. H.; Sheinin, Y.; Zhou, J.; Oupicky, D., Tumor-Penetrating Nanoparticles for Enhanced Anticancer Activity of Combined Photodynamic and Hypoxia-Activated Therapy. ACS nano 2017, 11, (2), 2227-2238.

2.     Sun, H.; Cao, D.; Liu, Y.; Wang, H.; Ke, X.; Ci, T., Low molecular weight heparin-based reduction-sensitive nanoparticles for antitumor and anti-metastasis of orthotopic breast cancer. Biomaterials science 2018, 6, (8), 2172-2188.

Point 4: Fig. 3, could the author show the images of IHC staining for E-cadherin, N-Cadherin, Vimentin, MMP2, and MMP9 in the lung tumor nodules to show the EMT phenotype in metastatic lesions with and without YFTL treatment?

Response 4: The images of IHC staining for E-cadherin, N-Cadherin, Vimentin, MMP2, and MMP9 in the lung tissues have been showed in Fig. 4.

Reviewer 2 Report

The manuscript entitled “Yifei Tongluo, a Chinese herbal formula, suppresses tumor growth and metastasis and exerts immunomodulatory effect in Lewis lung carcinoma mice” by Qi et al. describes an interesting study about the antitumor effects of the Yifei Tongluo herbal combination (YFTL) in Lewis lung adenocarcinoma mice. They demonstrated that YFTL decreases tumor burden and metastatic nodules, and increases survival in tumor-bearing mice; they also give an insight into the molecular mechanisms by analyzing several molecular markers of proliferation, angiogenesis, EMT and invasion both at the tumor and serum levels.

However, there are some aspects that should be improved:

1)      Lines 252-254: “However, in order to facilitate the translation of basic science to the clinical setting, further researches are needed to identify the specific ingredients in the YFTL granules that target the Akt/MAPK and TGF-β1/Smad2 pathways in the treatment of lung cancer.” As mentioned at the beginning of the manuscript, YFTL contain some components that have been demonstrated to possess antitumor properties. If this was known, and the authors mention the previous sentence in the discussion, why did they use the complete YFTL instead of a specific compound contained on it? Is there any advantage of using the herbal combination (YFTL)? This should be discussed.

2)      Immunomodulation by YFTL (T cells levels and NK cytotoxic activity): Of course the action of NK and T cells directed to the tumor is important to fight against the tumor, but a dramatic splenocyte proliferation (“dramatically promoted splenocytes proliferation”, line 168), in general, does it necessarily mean a positive effect or could it be considered as an adverse effect? The statement “These data suggest that YFTL 173 obviously improved the immunosuppressive state of Lewis tumor-bearing mice.” (line 173, and also mentioned in the abstract) should be reconsidered.

3)      Lines 249-252: “Consistent with these reports, our results illustrated that YFTL could suppress the angiogenesis and the activation of EMT via the downregulation of VEGF, N-cadherin, Vimentin, MMP-2 and MMP-9 and upregulation of the E-cadherin by suppression of Akt/ERK1/2 251 and TGF-β1/Smad2 pathways”- The results do not include enough evidences to suggest this, as there are not mechanistic studies.

4)      It would be interesting to discuss the potential combination of YFTL with the chemotherapy currently used in this cancer type.

5)      Lines 91-93: “There were two of 8 mice in the 91 MC group that lived to day 40 and all of them died by day 42, whereas all mice in the YFTL-H groups 92 lived to day 40”. It should be mentioned here what happened with group YFTL-L.

6)      Table 1: What does “ratio” mean here?

7)      Figure 4: Why were not E- and N-cadherin expression levels confirmed by WB?

8)      Figure 5: Were the levels of CD4+, CD8+ and NK cells also analyzed in YFTL-L mice? It would be worth to have the results to confirm whether there is a dose-dependent effect or not.

9)      The analysis performed to quantify IHC pictures (e.g. Figure 2) is not described. Please, include.

10)   The in vivo doses of YFTL in the text are not in accordance with the legend of Figure 1. Please check along the article.

11)   As the mouse tumor model is performed by injecting mouse lung adenocarcinoma cells (line LLC), it is not considered a xenograft.

Minor points:

12)   It would be better to mention before the M&M section, which is at the end, that YFTL is orally administered to mice.

13)   Figure 2: Scale bars are missing in some pictures. Letters in the legend are not in agreement with those in the figure.

14)   Figure 3: Scale bars are missing in some pictures.

15)   Figure 5: CD48-PE should be CD8-PE

Author Response

Response to Reviewer 2 Comments:

Point 1: Lines 252-254: “However, in order to facilitate the translation of basic science to the clinical setting, further researches are needed to identify the specific ingredients in the YFTL granules that target the Akt/MAPK and TGF-β1/Smad2 pathways in the treatment of lung cancer.” As mentioned at the beginning of the manuscript, YFTL contain some components that have been demonstrated to possess antitumor properties. If this was known, and the authors mention the previous sentence in the discussion, why did they use the complete YFTL instead of a specific compound contained on it? Is there any advantage of using the herbal combination (YFTL)? This should be discussed.

Response 1: The advantage of using the herbal combination (YFTL) was discussed in the Discussion section of revised manuscript as suggested. Although specific compounds in the YFTL possess antitumor properties, however our goal in this work was to utilize the synergetic effects of anti-proliferation, anti-resistance and immunomodulation in the YFTL combination. The review’s suggestion may be evaluated in future studies using the combination of individual compounds from the YFTL.

Point 2: Immunomodulation by YFTL (T cells levels and NK cytotoxic activity): Of course the action of NK and T cells directed to the tumor is important to fight against the tumor, but a dramatic splenocyte proliferation (“dramatically promoted splenocytes proliferation”, line 168), in general, does it necessarily mean a positive effect or could it be considered as an adverse effect? The statement “These data suggest that YFTL 173 obviously improved the immunosuppressive state of Lewis tumor-bearing mice.” (line 173, and also mentioned in the abstract) should be reconsidered.

Response 2: Previous studies have revealed that the tumor-bearing mice were in the state of immunosuppression. Compared with the model control mice, the proliferation of spleen lymphocyte was increased significantly in tumor-bearing mice but still lower than that in normal mice. So YFTL potentiated splenocyte proliferation that contributes significantly to its antitumor activity. Based on the results of the immune cell activity, cytokine expression and lymphocyte phenotype, we can conclude that YFTL showed immunomodulatory activities in improving the immunosuppressive state of tumor-bearing mice. Perhaps, the word “proliferation” used may be not accurate here and should be changed to “restoration”.

Point 3: Lines 249-252: “Consistent with these reports, our results illustrated that YFTL could suppress the angiogenesis and the activation of EMT via the downregulation of VEGF, N-cadherin, Vimentin, MMP-2 and MMP-9 and upregulation of the E-cadherin by suppression of Akt/ERK1/2 251 and TGF-β1/Smad2 pathways”- The results do not include enough evidences to suggest this, as there are not mechanistic studies.

Response 3: The sentence in Line 249-252 has been modified in the revised manuscript.

Point 4: It would be interesting to discuss the potential combination of YFTL with the chemotherapy currently used in this cancer type.

Response 4: The potential for combination of YFTL with the chemotherapy drugs was discussed in the Discussion section of the revised manuscript.

Point 5: Lines 91-93: “There were two of 8 mice in the 91 MC group that lived to day 40 and all of them died by day 42, whereas all mice in the YFTL-H groups 92 lived to day 40”. It should be mentioned here what happened with group YFTL-L.

Response 5: We have described the result of survival assay in the YFTL-L group in the revised manuscript.

Point 6: Table 1: What does “ratio” mean here?

Response 6: It means the ratio of each herbal medicine weight in the YFTL formula.

Point 7: Figure 4: Why were not E- and N-cadherin expression levels confirmed by WB?

Response 7: The expressions of E-cadherin N-cadherin, Vimentin, MMP-2 and MMP-9 were first tested by the immunohistochemical method. We believe that the assaying of Vimentin, MMP-2 and MMP-9 by Western Blot might be enough to confirm the immunohistochemical data.

Point 8: Figure 5: Were the levels of CD4+, CD8+ and NK cells also analyzed in YFTL-L mice? It would be worth to have the results to confirm whether there is a dose-dependent effect or not.

Response 8: The levels of CD4+ T cell, CD8+ T cell and NK cells were not analyzed since the high dose of YFTL was found to be more effective on the antitumor and anti-metastasis.

Point 9: The analysis performed to quantify IHC pictures (e.g. Figure 2) is not described. Please, include.

Response 9: The description of the analysis performed to quantify IHC pictures was added in the Materials and Methods section (4.5) of the revised manuscript.

Point 10: The in vivo doses of YFTL in the text are not in accordance with the legend of Figure 1. Please check along the article.

Response 10: The in vivo doses of YFTL in the text have been corrected in the revised manuscript.

Point 11: As the mouse tumor model is performed by injecting mouse lung adenocarcinoma cells (line LLC), it is not considered a xenograft.

Response 11: The “xenograft” has been removed in the revised manuscript as suggested.

Suggestions (Minor errors):

Point 12: It would be better to mention before the M&M section, which is at the end, that YFTL is orally administered to mice.

Response 12: We have added the administration of YFTL in the legend of Figure 1 and Results 2.1.

Point 13: Figure 2: Scale bars are missing in some pictures. Letters in the legend are not in agreement with those in the figure.

Response 13: In Figure 2, scale bars have been added in all pictures. The letters in the legend have been modified to match those in the figure.

Point 14: Figure 3: Scale bars are missing in some pictures.

Response 14: In Figure 3, the missing scale bars have been added in all pictures.

Point 15: Figure 5: CD48-PE should be CD8-PE

Response 15: In Figure 5, the “CD48-PE” has been changed to “CD8-PE” in the revised manuscript.

Reviewer 3 Report

The investigators have studied the activity of  Yifei Tongluo as an anti-cancer agent in the syngeneic Lewis lung model.  The manuscript is generally well constructed and detailed. The survival  benefit both YFTL doses are modest, but lack  of weight loss is good.

Figure 1A: Plotting tumor weights at a specific  time is far less convincing than showing tumor growth 0-21 days or at  least 0-14 days, where tumors are measured every day or every other day.  If the authors have this data it would be good  to show (Y-axis tumor size, X-axis days).

Figure 4. The impact on ‘EMT’ section: Vimentin in  immunoblots could be from macrophages/monocytes and stromal fibroblasts  and not from EMT-derived cells. This alternative should be mentioned in  the text.

Overall the data are interesting. In the Discussion  section the authors could comment on the potential for combinations of  YFTL with other standards of care in lung therapy. By itself the impact  of YFTL is modest and will likely require  adding to other agents (including checkpoint inhibitors) which might  add to the impact of the Discussion.

Author Response

Response to Reviewer 3 Comments:

Point 1: Figure 1A: Plotting tumor weights at a specific time is far less convincing than showing tumor growth 0-21 days or at least 0-14 days, where tumors are measured every day or every other day. If the authors have this data it would be good to show (Y-axis tumor size, X-axis days).

Response 1: The tumor weights were only measured at the end of the study in the Lewis lung model, which are plotted in Fig. 1B. Therefore, we could not construct a suggested plot of tumor weights and time. 

Point 2: Figure 4. The impact on ‘EMT’ section: Vimentin in immunoblots could be from macrophages/monocytes and stromal fibroblasts and not from EMT-derived cells. This alternative should be mentioned in the text.

Response 2: We have modified the text related to Figure 4 according to your suggestion.

Point 3: Overall the data are interesting. In the Discussion section the authors could comment on the potential for combinations of YFTL with other standards of care in lung therapy. By itself the impact of YFTL is modest and will likely require adding to other agents (including checkpoint inhibitors) which might add to the impact of the Discussion.

Response 3: We have discussed the potential for combination of YFTL with the chemotherapy drugs or checkpoint inhibitors in the Discussion section of the revised manuscript.

Round 2

Reviewer 1 Report

I am satisfied with the revision.